# Development of Multifunctional Coating of Textile Materials Using Silver Microencapsulated Compositions

**Luidmila Petrova, Olga Kozlova \*, Elena Vladimirtseva \*, Svetlana Smirnova \*, Anna Lipina and Olga Odintsova \***

Department of Chemical Technology of Fibrous Materials, Ivanovo State University of Chemistry and Technology, Sheremetevsky Ave., 7, 153000 Ivanovo, Russia; petrova_ls@isuct.ru (L.P.); prohorova.a94@yandex.ru (A.L.)

\* Correspondence: kozlova_ov@isuct.ru (O.K.); vladimirtseva_el@isuct.ru (E.V.); smirnova_sv@isuct.ru (S.S.); odintsova_oi@isuct.ru (O.O.)

**Abstract:** The efficiency of the method for the synthesis of silver nanoparticles using a system containing oxalic dialdehyde as a reducing agent, and polyguanidine as a stabilizer is shown. An analysis of the data of photon correlation spectroscopy characterizing the sizes of the formed particles in the Ag-polyelectrolyte system is presented. It has been established that the synthesized silver nanoparticles have a stable biocidal effect. The system of biodegradable polyelectrolytes chitosan-xanthan gum for the synthesis of the capsule shell including silver nanoparticles is selected. This will allow the formation of stable polyelectrolyte capsule shells containing oyster mushroom mycelium extract. A protocol for the synthesis of microcapsules by the method of sequential adsorption of chitosan polyelectrolytes and xanthan gum on calcium carbonate templates was developed. Silver nanoparticles are included in the capsule shell, and a biologically active drug (oyster mushroom mycelium extract) is included in the core. The technological mode of complex capsules immobilization on a textile material by the layer-by-layer method is described. The immobilization of multilayer microcapsules on a fibrous substrate is provided by a system of polyelectrolytes: positively charged chitosan and negatively charged xanthan gum. The developed multifunctional coatings make it possible to impart multifunctional properties to textile materials: antibacterial, antimycotic, high hygroscopic properties.

**Keywords:** cellulose textile material; microencapsulation; antibacterial; antimycotic; wound healing properties; silver; polyelectrolyte microcapsules

## 1. Introduction

Nanosilver is a universal biocide, the particles of which suppress the pathological effect of a wide range of microorganisms, including viruses. Modern methods for the formation of silver-containing composites in natural [1] and synthetic [2] matrices by direct impregnation of nanoparticles into the system, which exhibit high bacteriostatic (inhibition of bacterial growth) or bactericidal (destruction of inoculated bacteria) activity, are shown.

The new approach proposed by the authors is based on the inclusion of silver nanoparticles into the shell of a microcapsule containing a medicinal or biologically active compound.

Microencapsulation of drugs provides prolonged action and safety for humans due to the possibility of using minimum concentrations of the active substance and its controlled release [3]. The introduction of a biologically active or medicinal compound into the capsule core will make it possible to create medical textile materials with multifunctional properties: antibacterial, antimycotic, and wound healing.

Microencapsulation is widely used in various industries. In agriculture and in everyday life, microencapsulated insecticides are used, microcapsules with vitamins, essential and fatty oils are included in various cosmetics (creams, gels, serums), microencapsulated probiotics are used in feed and feed additives in veterinary medicine. The introduction of a new technology for microencapsulation of textile materials will make it possible to obtain a completely unique product with innovative quality and functional indicators.

Microencapsulation is the process of encapsulating a functional substance in a shell that protects it from evaporation, pollution, and the influence of other environmental influences and allows the substance to be released in a prolonged manner [4,5].

Depending on the thickness and material of the shell, the core contents can be released through changes in temperature or pH, biodegradation, etc., [6,7].

The existing microencapsulation methods are conventionally divided into chemical, physical [8], and physicochemical [9,10]. When choosing the most suitable method for each specific case, one proceeds from the specified properties of the final product, the cost of the process, and many other factors. However, the decisive factor is the properties of the encapsulated substance.

Physical methods of encapsulation include suspension crosslinking, solvent evaporation, simple and complex coacervation, phase separation, spray drying, fluidized bed spraying, melt crystallization, precipitation, co-extrusion, layering. Thus, the formation of capsules in this case occurs without chemical interaction [11–13].

Chemical methods of microencapsulation include emulsion polymerization, interfacial polymerization, dispersed and interfacial methods [14]. In this case, the shell materials can be monomers, oligomers or polymers having functional groups and capable of participating in reactions of growth or crosslinking of chains with the formation of high molecular weight linear and reticulated polymers.

Chemical methods also include the synthesis of polyelectrolyte nanocapsules, the "Layer-by-layer" method—electrostatic self-assembly, which was proposed by scientists of the Max Planck Institute in 1998 [15,16]. For the first time, the "Layer-by-layer" method was used to form monolayer ultrathin polymer films on a macroscopic substrate. In 1966, the authors [17] proposed to use sequential adsorption to assemble films. In 1991, Decher et al. considered a method for obtaining polyelectrolyte films, which consists in the alternate adsorption of polycations and polyanions on a substrate [18]. The synthesis of polyelectrolyte capsules consists in the deposition of oppositely charged polyelectrolytes on the surface of a solid particle called a template, which can be microparticles of polystyrene, silicon dioxide [19], calcium carbonate [20], cadmium carbonate. The capsule core is most often removed by dissolution. Its material affects the permeability of the capsule shell, its shape, morphology and the rate of removal of the core from the capsule. Calcium carbonate templates are the most frequently used in experiments, since in the case of using silicon dioxide, difficulties arise with the use of hydrofluoric acid, which is used to dissolve it. The use of polystyrene and melamine formaldehyde as templates is limited by their incomplete dissolution. When creating capsules synthesized for medical purposes, calcium carbonate is the most suitable because of its biocompatibility, biodegradability, as well as porous structure and large surface area, which allows it to be used for encapsulating various substances [21].

The formation of polyelectrolyte shells on colloidal particles of different nature is carried out by the method of alternate adsorption of oppositely charged polyelectrolyte macromolecules, as a result of which a shell of various thicknesses can be formed [22–24].

To form the capsule shell, synthetic (polystyrene sulfonate, polyacrylic acid, polydialyldimethylammonium chloride, etc.) and biocompatible polyelectrolytes (hylauronic acid, sodium alginate, chitosan, L-lysine, etc.) are used. The formation of polyelectrolyte shells occurs mainly through electrostatic interaction; hydrophobic interactions or the formation of hydrogen bonds can also occur [25–27].

The capsule core is most often removed by dissolution. When melamine-formaldehyde latex particles and tetrahydrofuran are used as templates [23,24,28], organic solvents are used.

Modification of microcapsules can be carried out in three ways: through the synthesis of nanoparticles in a polyelectrolyte shell, for example, gold nanoparticles [29–33], by incorporation into the core, or by adsorption of stabilized nanoparticles into a polyelectrolyte shell [34,35]. These methods are currently being developed and improved. The optical and antibacterial properties of silver nanoparticles are of great interest.

The analysis of the cited literature data showed a wide range of works of domestic and foreign scientists aimed at obtaining antibacterial materials for various purposes. Despite the large number of works, the problem of obtaining a stable release form of antibacterial drugs based on silver nanoparticles for finishing textile materials made from natural fibers and implementing the technology for their use remains not fully resolved. There are practically no technologies aimed at creating silver-containing microencapsulated antibacterial agents for processing textile materials for medical purposes.

One of the ways of imparting antimicrobial properties is the development of encapsulated antibacterial drugs and methods of their immobilization on textile materials. Microencapsulation methods make it possible to obtain particles of various sizes—from fractions of a micron to hundreds of microns.

Among such systems, we should especially note polyelectrolyte microcapsules (PEM), a significant property of which is the semi-permeability of the shell, which can allow small molecules to pass through, but retains high-molecular compounds.

This makes it possible to consider PEM as the main method of immobilization of proteins and high-molecular biologically active substances (BAS), when the semipermeable shell of the microcapsule separates the aqueous solution of the substrate from the protein solution and thus protects it from negative external influences. Additional advantages of polyelectrolyte capsules over other similar systems are their mono-dispersity with a wide range of specified sizes; simplicity of regulating their permeability and the possibility of a wide choice of shell material. The shells of such microcapsules can be modified, including various types of ions, functional molecules, nanoparticles.

The most acceptable from a practical point of view are two ways of encapsulating biologically active substances:

-nanoemulsion method;

-synthesis of nanocapsules using templates, where microparticles of calcium carbonate are used to form PEM.

Since the processed textile materials are planned to be used for medical purposes, it is necessary to use biocompatible biodegradable polyelectrolytes to form the capsule shell.

The purpose of this work is the formation of ultrathin multifunctional coatings on cellulose textile material using silver-containing microcapsules formed with the core of a wound-healing compound (oyster mushroom mycelium extract). The coatings are designed to create a cellulose material with high consumer characteristics, capable of inhibiting the vital activity of pathogenic bacteria and fungi, and at the same time exerting a wound-healing effect.

## 2. Materials and Methods

### 2.1. Materials

The object of the study was a cotton bleached calico of plain weave with a surface density of $142 \pm 7$ g/m$^2$, produced by JSC "Nordtex", Ivanovo.

The following polyelectrolytes were used: chitosan (manufactured by "Bioprogress" Ltd., Shchelkovo, Moscow, Russia) and xanthan gum (manufactured by "INGREDIKO" Ltd., Moscow, Russia).

### 2.2. Methods

The work uses a set of physical and chemical research methods (atomic absorption spectroscopy, dynamic light scattering, scanning electron microscopy), conventional and original methods for assessing mechanical strength of the fabric and special, including antimicrobial, consumer characteristics of textile materials, generally accepted and original methods for assessing the antimicrobial, antimycotic and consumer characteristics of textile materials. To determine the amount of atomic silver in the capsule shell, an MGA-915 atomic absorption spectrometer (Lumex Instruments Canada, Fraserview Place Mission, British Columbia, Canada) was used; SEM photographs were taken using a Solver 47 Pro, NT-MDT scanning atomic force microscope (NT-MDT, Zelenograd, Moscow, Russia).

### 2.3. Silver Nanoparticles Obtaining Procedure

The object of the study was an aqueous solution of silver nitrate $AgNO_3$ of analytical grade, provided by LLC Lenreaktiv, St. Petersburg. The concentration of silver in the investigated solutions varied from $0.24 \times 10^{-7}$ mol/dm$^3$ to $0.47 \times 10^{-4}$ mol/dm$^3$. An aqueous solution of glyoxal with a concentration of 0.1 to 2 mol/dm$^3$ was used as a reducing agent, which was prepared by introducing an appropriate sample into cooled bi-distilled water with continuous stirring using a magnetic stirrer. To synthesize silver nanoparticles, a reducing agent solution was added to a silver nitrate solution of a certain concentration. The ratio of the solutions was varied. The prepared solutions of silver nitrate with a reducing agent and a stabilizer were heated to a temperature of 30–90 °C for 5–60 min. The silver reduction reaction was carried out in air.

Determination of the size of silver particles in the studied hydrosols was carried out by dynamic light scattering on a Zetasizer Nano ZS device (Malvern Panalytical, Worcestershire, UK). The size of the synthesized silver nanoparticles used in the study is 2 nm. A sufficiently high uniformity of particle sizes of 96%–100% was achieved.

### 2.4. Microencapsulation Techniques for Functional Substances

2.4.1. Method of Obtaining Microparticles of Calcium Carbonate

Spherical colloidal particles of $CaCO_3$ were obtained by mixing solutions of $CaCl_2$ and $Na_2CO_3$ with a concentration of 0.15 M. The reaction mixture was stirred for 5 min under normal conditions. After the completion of the process, $CaCO_3$ particles were washed from $Na^+$ and $Cl^-$ ions with distilled water and evaporated. Measurements of the size of the obtained calcium carbonate particles and determination of the particle size distribution were performed by laser diffraction on an Analysette 22 NanoTec instrument (Fritsch GmbH, Idar-Oberstein, Germany).

2.4.2. Polyelectrolyte Capsule Shell Synthesis Procedure

To form a polyelectrolyte shell, we used calcium carbonate cores of various diameters and natural polyelectrolytes, for example, negatively charged polyelectrolyte xanthan gum and positively charged chitosan. Calcium carbonate cores have a negative surface charge, so a cationic polyelectrolyte was applied as the first layer. For this purpose, 100 mL of a polyelectrolyte solution with a concentration of 0.001 g/L was added to 0.5 g of cores. The suspension was stirred for 20 min, after which the particles were washed three times with water. The same procedure was carried out using an anionic polyelectrolyte solution. Thereafter, 10 mL of a colloidal solution of silver nanoparticles was added to the system. By the method of alternate adsorption of oppositely charged macromolecules on colloidal particles, a shell consisting of the required number of layers was obtained.

2.4.3. Dissolution of Carbonate Cores Method

The production of hollow polyelectrolyte shells—permeable capsules—was carried out by dissolving $CaCO_3$ cores with the addition of the trisodium salt of

ethylenediaminetetraacetic acid (EDTA). As a result, calcium is removed from the capsule due to the formation of a stable complex of this metal with EDTA. A 0.2 M aqueous solution of EDTA with a pH of 7.5 was poured into the capsule suspension and stirred for 20 min, then the suspension was washed three times with distilled water.

2.4.4. Determination of the Sensitivity of Microorganisms to Antimicrobial Drugs by the Disk Method (Diffusion Test)

The method is based on the suppression of the growth of microorganisms on a solid nutrient medium under the action of an antimicrobial drug applied to a paper or tissue disc. As a result of diffusion of the drug into the environment, a concentration gradient of the test drug is formed around the disc—a zone of suppression of the growth of microorganisms. The size of the growth inhibition zone determines the effectiveness of the drug in relation to the studied bacterial culture. Within certain limits, the diameter of the growth inhibition zone is inversely proportional to the minimum inhibitory concentration.

For the study, a suspension containing a standard number of viable cells was used, which was inoculated with a lawn on the surface of a dense nutrient medium—agar Giventalya–Vedminoy, which does not interfere with the diffusion of preparations into Petri dishes. The discs soaked in the test preparations are placed on the inoculation at a distance of 2.5 cm from the center of the dish in a circle. The samples are incubated under conditions favorable for each specific microorganism, in our case 20 h at 35 °C. Then the diameters of growth inhibition zones around the disc were measured in millimeters (taking into account the disc diameter). The area to be measured is the area where the growth of bacteria is completely absent.

2.4.5. Assessment of Wound Healing on Models of Excisional and Burn Wounds

To study the effectiveness of textile materials with microcapsules of biologically active substances and silver nanoparticles on wound healing, samples with four-layer capsules containing silver nanoparticles in various concentrations from 1.75 to 10.10 mg/L were taken. The studies were carried out on outbred rats for 28 days. Before creating excisional and burn skin wounds, the animals were anesthetized. After that, hair was removed on the dorsal side of the body of the rats in the region of the shoulder blades, and the skin was wiped with 70° ethanol. Then one part of the animals received full-thickness wounds with a diameter of 12–14 mm, the other—thermal burns by applying a metal coin heated for 1 min (100 °C) on the skin for 30 s. A bandage in the form of a piece of tissue measuring $2 \times 2$ cm$^2$ was applied to the wounds and fixed from the first day 2 times a day. Immediately after the creation of excisional wounds in some of the animals, the wounds were infected with the Staphylococcus aureus ATCC® 6538P™ strain at a dose of $1.5 \times 10^9$ cells/mL (10ME). The application of the drug in these animals was started in some rats from the first day, in others from the fifth day.

**3. Results**

*3.1. Protocol for the Synthesis of Biologically Active Substances Nanocapsules of Natural Origin*

At this stage of research, a method was developed for encapsulating water-soluble biologically active substances (for example, an oyster mushroom mycelium extract). The direct method of encapsulation was used [35,36], when the injection of a functional substance was carried out during the formation of templates. Spherical colloidal particles of $CaCO_3$ were selected as templates, which were obtained by mixing solutions of $CaCl_2$ and $Na_2CO_3$ with a concentration of 0.15–0.33 M. The reaction was expressed by the following equation:

$$CaCl_2 + Na_2CO_3 = CaCO_3 + 2NaCl \qquad (1)$$

When the solutions were quickly mixed, an amorphous precipitate of calcium carbonate was formed. After the completion of the process, the $CaCO_3$ templates were washed from $Na^+$ and $Cl^-$ ions with distilled water and evaporated.

To deposit polyelectrolyte layers on particles the method of polyion assembly was used. It was carried out by sequential treatment with oppositely charged polyelectrolytes. To form a polyelectrolyte shell, biodegradable polyelectrolytes were used: chitosan and xanthan gum. Since the calcium carbonate cores have a negative surface charge, a positively charged polyelectrolyte, chitosan, was applied as the first layer. To do this, 100 mL of a 0.1% polyelectrolyte solution prepared in the presence of 0.5 mol/l sodium chloride was added to 1.0 g of the cores. The suspension was stirred for 20 min. After adsorption of each polyelectrolyte layer, the suspension was centrifuged and the particles were washed three times with distilled water. Then the same procedure was carried out using a solution of negatively charged xanthan gum polyelectrolyte, concentration of 0.05%. Further, by the method of alternate adsorption of oppositely charged macromolecules on colloidal particles, it is possible to obtain a shell consisting of the required number of layers. In this work, two- and four-layer capsules were synthesized. At all stages of the experiment, the pH was varied in the range of 6.0–6.2.

Hollow polyelectrolyte shells—permeable capsules—were obtained by dissolving $CaCO_3$ cores with the addition of the trisodium salt of ethylenediaminetetraacetic acid (EDTA). The calcium carbonate from the capsule was removed by forming a stable calcium EDTA complex (see Section 2.4.3.)

By varying the conditions of the process (concentration of reagents, temperature, intensity of stirring of the reaction mixture, its duration), it is possible to obtain microspherolites with an average diameter of 2 to 11 μm and a fairly narrow size distribution.

To measure the size of the obtained calcium carbonate particles and determine the particle size distribution, the method of laser diffraction was used on an Analysette 22 NanoTec instrument. With an increase in the stirring time from 0.5 to 5 min, the size of the calcium carbonate particles obtained as a result of reaction (1) decreased from 8.68 to 6.96 μm (Table 1). Stirring the reaction mixture for more than 5 min did not lead to a change in the size of the synthesized particles.

**Table 1.** Influence of the mixing time of the reaction mixture on the dimensional characteristics of calcium carbonate templates.

| Indicators | Value of Indicators | | | | | | |
|---|---|---|---|---|---|---|---|
| Stirring time of the reaction mixture, min | 0.5 | 1.5 | 3 | 4 | 5 | 6 | 10 |
| Dimensional characteristics of $CaCO_3$ particles, μm | 8.68 | 8.6 | 7.77 | 7.68 | 6.96 | 6.96 | 6.96 |

Oyster mushroom mycelium extract was chosen as a BAS. Basidiomycete Pleurotus ostreatus (oyster mushroom) is widely used in medical practice as producers of various drugs [37]. Extracts obtained from its mycelium are capable of inhibiting the growth of malignant tumors and, at the same time, exhibiting strong immunostimulating activity [38,39]. They are complex systems that include various biologically active components, namely: ubiquitin—like proteins, lectins, proteases, and glucans with multifunctional biomedical activity.

The permeability of the polyelectrolyte shell of a microcapsule can be changed by varying the pH rage [40]. The method is based on shielding the electrostatic interaction between polyelectrolytes, while an increase in the diameter of microcapsules is observed with a sequential change in the pH range from alkaline to acidic. Therefore, for the introduction of biologically active substances (oyster mushroom mycelium extract) inside the microcapsules, it is necessary to acidify the solution. This contributes to the partial "loosening" of the microcapsules, in which pores are created in this case, this allows the

BAS to penetrate into the capsule. For this, the system was added 2–5 mL of 3% acetic acid solution to pH = 3.5–4. Then, with stirring, 100 μL of BAS was introduced. After vigorous stirring for 15–20 min, neutralization was carried out with sodium hydroxide solution (1%) to pH = 6. The suspension was centrifuged, the upper fraction of the supernatant was discarded, and distilled water was added.

Based on the developed technique, capsules with di-, tri-, and tetraloid shells, including a BAS solution, were synthesized.

To confirm the presence of capsules in the system under study by dynamic light scattering, the sizes of synthesized capsules were measured on a Photocor Compact-Z device (photon correlation spectroscopy) and photographs were obtained using a Mikmed-6 microscope with a camera.

Table 2 shows various protocols for the preparation of polyelectrolyte shells on calcium carbonate particles.

**Table 2.** Influence of the capsule formation technique on their appearance and aggregate stability.

| No. | Photographs of Particles Obtained Using a Microscope "Micromed 1" | Method for Obtaining Capsules | Particle Size Distribution | |
| | | | Particle Size, nm | Percentage, % |
| --- | --- | --- | --- | --- |
| 1 |  | 1.1 1.0 g of calcium carbonate particles were treated for 20 min in a solution of chitosan with a concentration of 0.1%; - rinsing with water; - impregnation with a solution of xanthan gum 0.05% 20 min; - rinsing with water; 1.2. dissolution of the core. | 9.7 316.1 7597.0 | 0.1 24.6 75.3 |
| 2 |  | 2.1. as in item 1.1 2.2. chitosan layer (according to the technology above); -rinsing with water; 2.3. dissolution of the core; 2.4. acidification and injection of 100 μL BAS; -stirring and neutralization with sodium hydroxide. | 496.0 909.7 7989.0 | 0.4 78.5 21.1 |
| 3 |  | The lifetime of the system prepared according to the developed method is 6 days. | 522.0 922.8 7998.1 | 2.0 97.2 0.8 |

The first sample, as can be seen from the photograph, are hollow capsules, the contents of which are completely transparent. The capsule size was 7000–8000 nm. The capsules presented in the second sample were synthesized according to the developed method. As can be seen from the photograph, the capsule core has a dark color, which corresponds to the color of the BAS injected into the capsule. It can be argued that the technology of forming the shell allows purposefully change of the permeability of the capsule shell by varying the pH of the suspension, thereby ensuring the penetration of biologically active substances into the core. The final size of the capsule had changed to 900–1000 nm (70%), at the same time, large capsules remained, about 8000 nm (20%), and a small number of capsules of 496–522 nm appeared.

The aggregation of the capsules was monitored using an optical microscope. The capsules synthesized according to the developed protocol in the system tend to the formation of associates, however, no associates were observed within a week (Table 2, sample 3). Since the technology involves the application of freshly prepared microencapsulated compositions to the textile material, the stability of the suspension for 6 days is sufficient. Using a scanning electron microscope, photographs of microcapsules applied to the textile material were obtained (Figure 1).

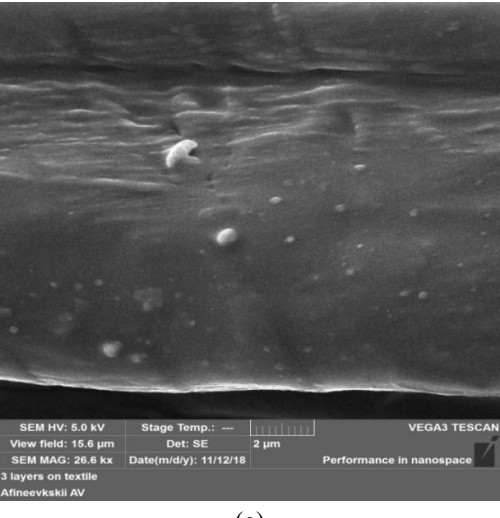
(**a**)

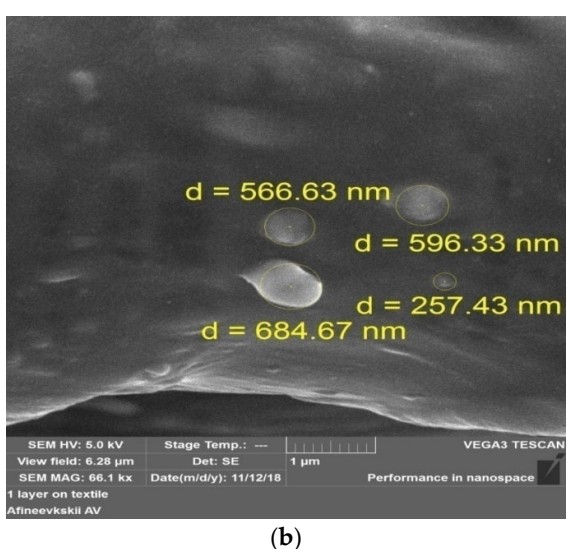
(**b**)

**Figure 1.** SEM photograph of textile material containing polyelectrolyte microcapsules.

### 3.2. Atomic Adsorption Microscopy of Microcapsules Containing Silver Nanoparticles

To provide the capsule shell bactericidal properties, a technology was developed for immobilizing silver nanoparticles into its composition. The idea was that the preparations applied to the textile material first showed a bactericidal effect, disinfecting the wound, and then the released medicinal substance had a directed effect. For this, a positive polyelectrolyte, chitosan, was applied to the calcium carbonate core in the first layer, washed three times with distilled water, and a layer of negative polyelectrolyte, xanthan gum, was applied in the second layer and washed with water. Before applying the third polyelectrolyte layer consisting of positively charged chitosan, microparticles were impregnated in a solution of silver nanoparticles.

The use of atomic absorption spectroscopy made it possible to determine the presence of silver nanoparticles in the microcapsules. It was shown that the amount of silver in the nanoform per gram of microcapsules was 0.0067 mg.

The resulting capsules were applied to the textile material according to the following process: impregnation with a solution of chitosan with a concentration of 5 g/L, drying at 1000 °C for 3 min, impregnation with a nanodispersion of the encapsulated preparation, drying at a temperature of 100 °C for 3 min, after which the antibacterial activity of the

samples was determined in relation to gram positive and gram-negative microorganisms and fungi (Table 3.).

**Table 3.** Antibacterial activity of the obtained samples in relation to Staphylococcus aureus, Escherichia coli, and Candida albicans.

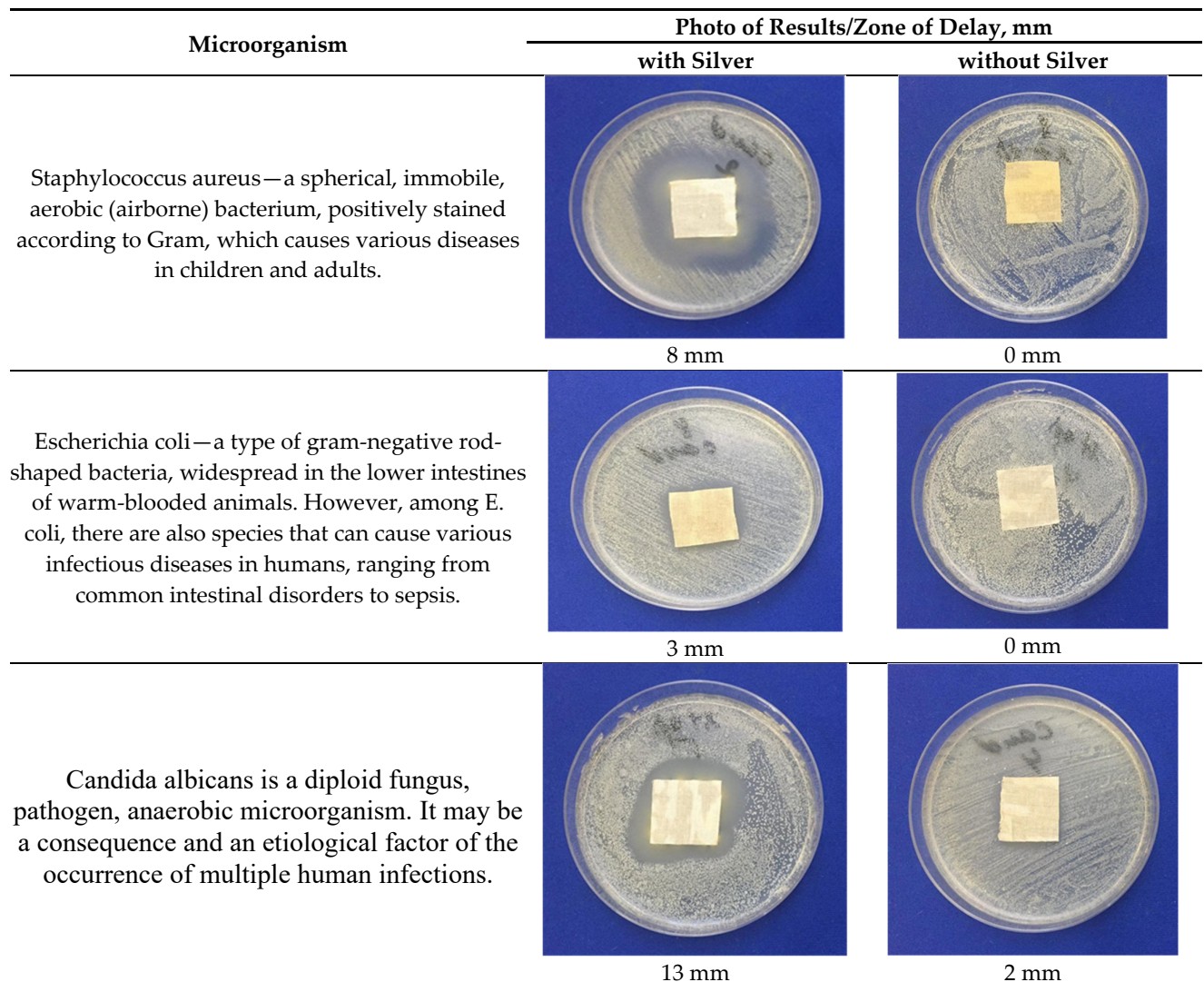

| Microorganism | Photo of Results/Zone of Delay, mm | |
| --- | --- | --- |
| | **with Silver** | **without Silver** |
| Staphylococcus aureus—a spherical, immobile, aerobic (airborne) bacterium, positively stained according to Gram, which causes various diseases in children and adults. | 8 mm | 0 mm |
| Escherichia coli—a type of gram-negative rod-shaped bacteria, widespread in the lower intestines of warm-blooded animals. However, among E. coli, there are also species that can cause various infectious diseases in humans, ranging from common intestinal disorders to sepsis. | 3 mm | 0 mm |
| Candida albicans is a diploid fungus, pathogen, anaerobic microorganism. It may be a consequence and an etiological factor of the occurrence of multiple human infections. | 13 mm | 2 mm |

The obtained results confirmed the antibacterial efficiency of the developed chemical product.

### 3.3. Functional Characteristics of Finished Textile Materials

The hydrophilic properties of the treated samples of textile materials were monitored by such indicators as capillarity, wettability, and water absorption, determined in accordance with standard methods [41]. It was found that before and after the application of the antibacterial coating, the capillarity remains at the level of 150–155 mm, the wettability is less than 1 s, the water absorption is −20–20.2 g/m$^2$, which fully meets the requirements for these materials.

The reparative effect of textile materials with a multifunctional coating was investigated on outbred rats for 28 days according to the following criteria:

- local inflammatory response;
- results of planimetric studies;

- timing of wound healing;
- determination of the index of acceleration of healing in the experimental groups in relation to self-healing wounds in the control groups.

During the experiments, all animal welfare standards were observed

Evaluation according to these criteria was carried out on 4, 8, 11, 15, 18, and 22 days after the start of treatment.

All wounds healed the same way. No inflammatory processes were registered. Planimetric survey data are presented in Table 4.

**Table 4.** Results of planimetric studies of wounds.

| Model | The Beginning of Wound Treatment | Drug Concentration, mg/L | Number of Animals | Percentage of Healing Relative to the Primary Wound,% | | | | | | Healing Acceleration Index,% |
|---|---|---|---|---|---|---|---|---|---|---|
| | Day from Model Creation | | | 4 | 8 | 11 | 15 | 18 | 22 | - |
| Burn wound | From the first day | - | 1 | 0 | 0 | 0 | 23.1 | 38.5 | 53.9 | - |
| | | 1.75 | 1 | 0 | 0 | 0 | 7.6 | 30.8 | 46.2 | −14.3 |
| | | 3.84 | 1 | 0 | 0 | 0 | 0 | 30.8 | 46.2 | −14.3 |
| | | 5.93 | 2 | 0 | 0 | 0 | 11.6 | 29.7 | 44.6 | −17.3 |
| | | 8.01 | 2 | 0 | 0 | 0 | 14.3 | 27.4 | 56.6 | +5.0 |
| | | 10.1 | 2 | 3.6 | 3.6 | 21.4 | 26.2 | 42 | 57.1 | +5.9 |
| Excision wound | From the first day | - | 1 | 0 | 35.7 | 57.1 | 85.7 | 85.7 | 85.7 | - |
| | | 1.75 | 1 | 0 | 33.3 | 50.0 | 66.7 | 75,0 | 83.3 | −2.8 |
| | | 3.84 | 1 | 0 | 41.7 | 66.7 | 83.3 | 91.7 | 91.7 | +7.0 |
| | | 5.93 | 2 | 3.6 | 47.8 | 64.0 | 73.1 | 77.6 | 92.3 | +7.7 |
| | | 8.01 | 2 | 0 | 28.7 | 48.7 | 74.3 | 87.6 | 100 | +16.7 |
| | | 10.1 | 2 | 0 | 45.0 | 55.8 | 78.3 | 90.0 | 100 | +16.7 |
| Excision wound + staphylococcus | From the first day | - | 2 | 0 | 46.5 | 60.7 | 75.0 | 92.9 | 92.9 | - |
| | | 1.75 | 1 | 7.1 | 42.9 | 71.4 | 78.6 | 85.7 | 92.9 | 0 |
| | | 3.84 | 2 | 9.8 | 46.9 | 63.4 | 89.8 | 93.2 | 96.9 | +4.3 |
| | | 5.93 | 2 | 0 | 30.8 | 61.4 | 80.8 | 84.6 | 92.3 | −0.6 |
| | | 8.01 | 2 | 0 | 37.5 | 62.6 | 93.8 | 100 | 100 | +7.6 |
| | From the fifth day | 1.75 | 1 | 0 | 30.8 | 38.5 | 69.2 | 84.6 | 92.3 | −0.7 |
| | | 3.84 | 1 | 0 | 40.0 | 60.0 | 66.7 | 100 | 100 | +7.6 |
| | | 5.93 | 2 | 0 | 37.3 | 61.4 | 92.3 | 92.3 | 96.2 | +3.6 |
| | | 8.01 | 2 | 0 | 25.0 | 31.3 | 43.8 | 56.3 | 75.1 | −19.2 |

The results obtained showed that only materials with microencapsulated drug concentrations of more than 5.93 mg/L had a positive effect on the rate of healing of burn wounds. Materials with any of the suggested concentrations had a positive effect on excision wounds, up to 17% on uninfected ones, and up to 7.5% on infected ones.

Application of the Staphylococcus aureus ATCC 6538P strain to wounds at a dose of $1.5 \times 10^9$ cells/mL (10ME) did not lead to wound infection (except for one). However, this method of creating purulent wounds, described in the literature, requires improvement. The result of the reparative action of textile materials with biologically active substances on such wounds is ambiguous.

## 4. Conclusions

A technology for the preparation of capsules by the method of sequential adsorption of polyelectrolytes of chitosan and xanthan gum, the shells of which included silver nanoparticles, and the core was a wound-healing agent was developed. The capsule shell synthesized using a polyelectrolyte assembly consisted of biodegradable and safe natural polymers. Nano-silver located between the polyelectrolyte layers gave the finish an antimicrobial effect. In this study, oyster mushroom mycelium was successfully used as a model filling of the capsule core. The proposed model of the formation of microcapsules,

the shells of which are doped with silver nanoparticles, allows the most targeted effect on the focus of infection, providing an antimicrobial effect. The encapsulated preparation of the oyster mushroom mycelium extract was released for a long time and provided in this case a more effective wound healing effect.

Cellulose tissue samples coated with silver-containing microcapsules have demonstrated high antibacterial, antimycotic activity and wound healing properties.

It was found that the optimal concentration of the microencapsulated drug for effective wound healing is 1%–2% of the weight of the textile material. In the case of burn wounds, to restore the skin, the material should be changed at least after two weeks.

Changing the filler of the capsule core (biologically active substance) makes it possible to impart a spectrum of different properties to the developed drug, and to the textile material—various types of final finishing.

**Author Contributions:** conceptualization, methodology, writing—preparation of the initial draft, writing—review and editing, O.O. and O.K.; data curation, research, visualization L.P. and A.L.; software, investigation, visualization, S.S. and E.V. All authors have read and agreed to the published version of the manuscript

**Funding:** This work was financially under the agreement with LLC "Smart Textile" No. 09.121.18., Carried out under the HealthNet program of the Innovation Promotion Fund.

**Institutional Review Board Statement:** The ethical review and approval of this study was rejected due to the lack of a need for this procedure in the Russian Federation. There is no independent law on the protection of experimental animals. Experiments on living organisms are regulated by order of the USSR Ministry of Health No. 755 of August 12, 1977 "On measures to further improve organizational forms of work using experimental animals." All experiments were carried out according to the above order.

**Informed Consent Statement:** Informed consent was obtained from all subjects involved in the study.

**Data Availability Statement:**  The data presented in this study are available on request from the corresponding author. The data is not publicly available due to the lack of patents. In the near future, patents will be issued and obtained, and data will become publicly available.

**Acknowledgments:** The authors would like to express their gratitude to Zacharov S.V. for his help in the microbiological studies and assessment of the healing properties of the samples (LLC Inbiopharm). The work was done on the equipment of the Center for Collective Use of Ivanovo State University of Chemistry and Technology.

**Conflicts of Interest:** The authors declare no conflicts of interest.

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
