# Peer review of "Development of Multifunctional Coating of Textile Materials Using Silver Microencapsulated Compositions"

_coatings, doi:10.3390/coatings11020159_

Round 1

Reviewer 1 Report

The manuscript (coatings-1074763) illustrates the preparation of textiles materials functionalized with hollow polyelectrolyte shell microcapsules. The microcapsules were loaded with biologically active substances (Oyster mushroom mycelium extract) at the interior and silver nanoparticles were embedded in the polyelectrolyte shell.

The hollow interior was generated through an extensively used technique involving the use of CaCO3 as a templating agent. The polyelectrolyte shell was constructed using chitosan as a positive layer and xanthan gum as a negative layer employing a layer-by-layer deposition technique. The Ag nanoparticles were included in the polyelectrolyte layers.

On the whole, the manuscript is fairly well-written and logically arranged. The research topic tackled – antimicrobial, antimycotic textiles are currently of great interest. The results are informative. Nevertheless, I would recommend the publication of this paper in Coatings (ISSN 2079-6412) on the condition that the manuscript undergoes some revisions and the following points will be taken into consideration.

Comments:

  1. The novelty of the study should be better addressed in the Introduction section. The authors should better highlight the originality of their approach and what benefit or new insights does it bring.
  2. Materials and methods section must be improved to include: 
  3. detailed information about characterization equipment used. The authors should present the equipment type (for example SEM, DLS should be mentioned here not in the results section – there only the results and discussion should be presented and not the procedures).
  4. synthesis procedure for Ag nanoparticles is missing completely, the only information is given in the abstract section.
  5. fabrication technique or transfer technique of the microcapsules to the textiles is also missing here - it is found however in the results section, but some corrections should be made, for example, line 306 page 8 indicates 1000°C).

In conclusion section 2 needs further input and rearrangement of the presentation with information taken from the result section.

  1. Results section:
  2. Table 1 – why does the particle size decrease with the increase in stirring time, is it an actual particle decrease (particle erosion) or just a larger particle size distribution caused by a more complete precipitation process? Since longer times do not change particle size, I consider not particle erosion. DLS distribution analysis should clarify this and could be used for discussion of the DLS results after the generation of the polyelectrolyte shell. Further, zeta potential measurements of the final microcapsules should bring information about their stability.
  3. Table 2 – the images presented have no scale bar, so they are irrelevant. Again, DLS should be preferred for evaluation of size distribution and SEM analysis for individual particle morphology (could highlight the porous - hollow characteristics)
  4. Figure 2 – scale bars needed. How can be explained the particle size on the textile? Does a particle size contraction occur during the drying stage of the deposition on the textile technique or the analysis presents the smallest fraction of the capsules? Better images should be selected!
  5. The adhesion characteristics of the polymer particles to the textiles should also be investigated. Is there a risk that a portion of the particles migrates especially in environments such as wound dressing and long contact time?
  6. The conclusion section should also be correlated with the novelty and better highlight the original and novel contribution this study brings forward.

Author Response

Dear Reviewer!

The authors thank you for the work done and for recommending the article for publication.

All comments have been carefully reviewed and revisions have been made.

Further you can read the authors answers.

1.The novelty of the study should be better addressed in the Introduction section. The authors should better highlight the originality of their approach and what benefit or new insights does it bring.

Response 1: The authors agree with the remark. The novelty of the research is reviewed and supplemented in the section "Introduction" - p. 1, lines 27-40.

2.Materials and methods section must be improved to include: 

3.detailed information about characterization equipment used. The authors should present the equipment type (for example SEM, DLS should be mentioned here not in the results section – there only the results and discussion should be presented and not the procedures).

Response 2,3: Information about the equipment added to sections 2.2., 2.3. and 2.4.– p. 4, lines 148-169, 175-178, 187-188.

4.synthesis procedure for Ag nanoparticles is missing completely, the only information is given in the abstract section.

Response 4: An Ag nanoparticle synthesis procedure added in section 2.3 - page 4, lines 155-169.

5.fabrication technique or transfer technique of the microcapsules to the textiles is also missing here - it is found however in the results section, but some corrections should be made, for example, line 306 page 8 indicates 1000°C).

Response 5: Changes made according to comments - page 9, line 339. Temperature has been corrected.

1.Results section:

2.Table 1 – why does the particle size decrease with the increase in stirring time, is it an actual particle decrease (particle erosion) or just a larger particle size distribution caused by a more complete precipitation process? Since longer times do not change particle size, I consider not particle erosion. DLS distribution analysis should clarify this and could be used for discussion of the DLS results after the generation of the polyelectrolyte shell. Further, zeta potential measurements of the final microcapsules should bring information about their stability.

Response 1,2: Table 1 shows the stirring time of the reaction mixture, into which the active reagent was constantly introduced with stirring. We fully agree with the opinion of the reviewer, studies to identify possible particle erosion will be carried out in the future.

Information on capsule stability has been received but is under discussion.

3.Table 2 – the images presented have no scale bar, so they are irrelevant. Again, DLS should be preferred for evaluation of size distribution and SEM analysis for individual particle morphology (could highlight the porous - hollow characteristics)

Response 3: The scale bar in the images presented in Table 2 has been added - p. 7-8.

4.Figure 2 – scale bars needed. How can be explained the particle size on the textile? Does a particle size contraction occur during the drying stage of the deposition on the textile technique or the analysis presents the smallest fraction of the capsules? Better images should be selected!

Response 4: The scale bar in Fig. 2 added - page 9. During the drying process, a decrease in the size of the capsules applied to the textile material was observed. At the moment, the authors are studying this process.

5.The adhesion characteristics of the polymer particles to the textiles should also be investigated. Is there a risk that a portion of the particles migrates especially in environments such as wound dressing and long contact time?

Response 5: High adhesion of chitosan-based polymer particles is known from literature sources (Roy, Jagadish & Salaün, Fabien & Giraud, Stéphane & Ferri, Ada & Guan, Jinping. (2017). Chitosan-Based Sustainable Textile Technology: Process, Mechanism, Innovation, and Safety. 10.5772 / 65259).  Chitosan-based particles have medicinal properties and can migrate during prolonged contact with the provision of additional medicinal effect.

6.The conclusion section should also be correlated with the novelty and better highlight the original and novel contribution this study brings forward.

Response 6: The necessary amendments have been made to the conclusion section - p. 12, lines 385-393.

The authors express their deep gratitude to the reviewer for the attention shown, the thorough study of the article material and the correct comments, with which they completely agree.

Reviewer 2 Report

Recommendation: I recommend the manuscript “Development of multifunctional coating of textile materials using microencapsulated silver compositions ” by Luidmila Petrova, Olga Kozlova, Elena Vladimirtseva, Svetlana Smirnova, Anna Lipina and Olga Odintsova for publication in the Coatings journal.

The manuscript provides valuable insights into the new type of materials composed of textiles coated with microcapsules containing bioactive agents and testing them as materials for medical applications. In the manuscript, two types of bioactive agents have studied Oyster mushroom mycelium extract and silver nanoparticles. They have been encapsulated in polyelectrolytes microparticles, and the textiles were coated with them via adsorption process. Antibacterial and antimycotic activity of such textiles has been analyzed. However, the bioactivity of this material is not proved efficiently by the studies presented in the manuscript. Several studies/measurements should be added to show the biocidal activity of the prepared material. Furthermore, the manuscript should, in some places, be improved.

Comments:

  1. My main criticism is that there is a lack of full description of the materials' syntheses considered in the manuscript. Silver nanoparticles’ synthesis is not given. Some particle characterization methods are only listed in the methodology section, but results are not presented and discussed (DLS. Mechanical characterization). Additionally, silver nanoparticles average size; its distribution they are not given. It is not sure whether we have silver nanoparticles inside these capsules, or maybe there is a bulk silver??? Authors don’t give any proofs. ASA gives only silver concentration.
  2. In Kirby-Bauer disc susceptibility assay a control sample should be tested, it means without silver nanoparticles to see whether the microcapsules alone exhibit antibacterial activity (see RSC Adv., 2015, 5, 58403–58415, this reference should be added)
  3. The Authors should try dynamic shake flask method test (DSFM) which is commonly used for textiles and composite materials to verify the antimicrobial and antimycotic activity of additives (see Acta Biomater. 5 (6) (2009) 2279–2289; Nanomaterials 2020, 10 (11), 2245, these references should be discussed and added)
  4. The name of reducing agent oxalic acid dialdehyde is not correct; glyoxal or oxalic dialdehyde are used for this compound.
  5. Table 1 and Table 4 are unreadable, a form of the presentation of the results must be changed.
  6. Particles’ size distribution (for example from DLS) should be presented.
  7. It is unclear whether the textiles were impregnated using the particles containing silver nanoparticles and Oyster mushroom mycelium extract or only AgNPs; it should be explained.
  8. It is not clear how these percentage results in Table 4 were calculated. It should be explained.

Author Response

Dear Reviewer!

The authors thank you for the work done.

All comments have been carefully reviewed and revisions have been made.

Further you can read the authors answers.

1.My main criticism is that there is a lack of full description of the materials' syntheses considered in the manuscript. Silver nanoparticles’ synthesis is not given. Some particle characterization methods are only listed in the methodology section, but results are not presented and discussed (DLS. Mechanical characterization). Additionally, silver nanoparticles average size; its distribution they are not given. It is not sure whether we have silver nanoparticles inside these capsules, or maybe there is a bulk silver??? Authors don’t give any proofs. ASA gives only silver concentration.

Synthesis of silver nanoparticles added in section 2.3, page 4, lines 155-169. Particle sizes determined by the DLS method are given in the article in Table 2, pages 7-8, lines 304-313. Silver nanoparticles are introduced between polyelectrolyte layers during the formation of the capsule shell. Section 2.4.1.

2.In Kirby-Bauer disc susceptibility assay a control sample should be tested, it means without silver nanoparticles to see whether the microcapsules alone exhibit antibacterial activity (see RSC Adv., 2015, 5, 58403–58415, this reference should be added)

Control data has been added to Table 3, pages 9-10. At the moment, the authors do not have the financial ability to purchase access to this article. Based on this, the authors do not have the right to give a link to an article they have not studied.

3.The Authors should try dynamic shake flask method test (DSFM) which is commonly used for textiles and composite materials to verify the antimicrobial and antimycotic activity of additives (see Acta Biomater. 5 (6) (2009) 2279–2289; Nanomaterials 2020, 10 (11), 2245, these references should be discussed and added)

The authors have read the articles and are grateful for the interest and help of the reviewer. In further research, the method will certainly be tested.

4.The name of reducing agent oxalic acid dialdehyde is not correct; glyoxal or oxalic dialdehyde are used for this compound.

The authors agree with the remark. Changes applied.

5.Table 1 and Table 4 are unreadable, a form of the presentation of the results must be changed.

The tables have been modified, changes have been made.

6.Particles’ size distribution (for example from DLS) should be presented.

Particle distribution was added to Table 2.

7.It is unclear whether the textiles were impregnated using the particles containing silver nanoparticles and Oyster mushroom mycelium extract or only AgNPs; it should be explained.

Microcapsules containing silver nanoparticles in the shell and oyster mushroom mycelium extract in the core were applied to textile materials.

8.It is not clear how these percentage results in Table 4 were calculated. It should be explained.

The calculation was carried out by biologists of Inbiopharm LLC.

The authors express their deep gratitude to the reviewer for the attention shown, the thorough study of the article material and the correct comments, with which they completely agree.

Reviewer 3 Report

  1. The title suggests multifunctionality of coating but only antimicrobial effectiveness is presented. Therefore the title has to be altered of more data included in the paper.
  2. Abstract - Raw 16 biologically active drug should be specified
  3. Abstract Raw 21 mentions UV-protection and wear resistance, but the results are not included in the paper. It is only the assumption that functionality of UV-protection can be achieved.
  4. Methods - Solution of Ag nanoparticles is not defined, nor the procedure of Ag microcapsules formation. 
  5. Raw 166 - is applied disk method a standard one? Why was ISO 20743 standard for antimicrobial testing not used? It is unclear how many samples were used for one test (triplicate or more) and what is the standard deviation of presented results (3, 8 and 13mm).
  6. Raw 290 - Fig 2 - data on applied magnification is missing! 
  7. Raw 310 - Fig 3 - description of samples is missing. It is unclear which contentration of nanoparticles was applied? In raw 189 different range of concentrations is mentioned for assesment of would healing. Please present the same range (1,75 - 10,0 mg/L) for antimicrobial efficiency as well.
  8. Only Staphilococcus bacteria is defined. Please define ATCC strain for Echerichia and Candida as well.
  9. Raws 343 and 344. This sentence is not very clear. Please explain in more details what is ambiguous.

Author Response

Dear Reviewer!

The authors thank you for the work done.

All comments have been carefully reviewed and revisions have been made.

Further you can read the authors answers.

1.The title suggests multifunctionality of coating but only antimicrobial effectiveness is presented. Therefore the title has to be altered of more data included in the paper.

The multifunctionality of the coating is determined by a complex of properties: wound healing, antibacterial and antimycotic characteristics.

2.Abstract - Raw 16 biologically active drug should be specified

Specification about the biologically active drug used (oyster mushroom mycelium) has been made (Abstract - Raw 17)

3.Abstract Raw 21 mentions UV-protection and wear resistance, but the results are not included in the paper. It is only the assumption that functionality of UV-protection can be achieved.

The authors obtained data on UV protection and wear resistance, but the results are still being discussed. These statements were excluded from the article.

4.Methods - Solution of Ag nanoparticles is not defined, nor the procedure of Ag microcapsules formation. 

The procedure for the formation of silver nanoparticles was added in section 2.3., Page 4, lines 155-169, the composition of microcapsules with a silver content in the shells is described on page 9.

5.Raw 166 - is applied disk method a standard one? Why was ISO 20743 standard for antimicrobial testing not used? It is unclear how many samples were used for one test (triplicate or more) and what is the standard deviation of presented results (3, 8 and 13mm).

The disc method is a simple and versatile method that allows you to qualitatively determine the antibacterial activity of the compounds under study. The required technical equipment is not available for the ISO standard method. The authors hope, through the publication of this article, to find like-minded people and possible funding for further research. We used 5 parallel samples, the standard deviation is 0.3 mm.

6.Raw 290 - Fig 2 - data on applied magnification is missing! 

Data added - fig. 1 page 9.

7.Raw 310 - Fig 3 - description of samples is missing. It is unclear which contentration of nanoparticles was applied? In raw 189 different range of concentrations is mentioned for assesment of would healing. Please present the same range (1,75 - 10,0 mg/L) for antimicrobial efficiency as well.

Data added. The article discusses solutions with one concentration ranking - 1.75-10.1 mg / L. Row 221 says a concentration of 1.75-10.0, as in Table 4 (concentration of 1.75 to 10.1).

8.Only Staphilococcus bacteria is defined. Please define ATCC strain for Echerichia and Candida as well.

Data were presented in table 3.

9.Raws 343 and 344. This sentence is not very clear. Please explain in more details what is ambiguous.

In this case, the authors had in mind that the results obtained with infection of wounds with staphylococcus aureus are incorrect and the technique requires improvement.

The authors express their deep gratitude to the reviewer for the attention shown, the thorough study of the article material and the correct comments, with which they completely agree.

Reviewer 4 Report

The paper “Development of multifunctional coating of textile materials using silver microencapsulated compositions”, by Luidmila Petrova et al., present interesting results concerning wound healing by using silver microencapsulated compositions. The paper is very well written, and the information is clearly presented.

 The paper is recommended for publication with minor revisions.

  1. As a general remark: the authors should use subscript for the chemical formulas.
  2. In Table 1: the values presented on the right-hand side are not so clear. Please insert additional space between the values, or change the table representation.
  3. In Table 2: the authors are asked to insert the scale on each optical image.
  4. In Figure 2: again, please insert the scale on the SEM images.
  5. In the paper you talked about wound healing for rats, but I didn’t saw any ethical statement in the paper. Usually when are experiments performed on animals, you should insert a statement, etc.

Author Response

Dear Reviewer!

The authors thank you for the work done and for recommending the article for publication.

All comments have been carefully reviewed and revisions have been made.

Further you can read the authors answers.

1.As a general remark: the authors should use subscript for the chemical formulas.

Corrected according to comments

2.In Table 1: the values presented on the right-hand side are not so clear. Please insert additional space between the values, or change the table representation.

Corrected according to comments (p. 7)

3.In Table 2: the authors are asked to insert the scale on each optical image.

Corrected in accordance with the comments - in table 2 added a scale - p. 7.

4.In Figure 2: again, please insert the scale on the SEM images.

Corrected in accordance with the notes - in fig. 2, a scale was inserted into the SEM images. (p. 9)

5.In the paper you talked about wound healing for rats, but I didn’t saw any ethical statement in the paper. Usually when are experiments performed on animals, you should insert a statement, etc.

Additions made to article page 11, line 362.

The authors express their deep gratitude to the reviewer for the attention shown, the thorough study of the article material and the correct comments, with which they completely agree.

Round 2

Reviewer 2 Report

Recommendation: I recommend the manuscript “Development of multifunctional coating of textile materials using microencapsulated silver compositions ” by Luidmila Petrova, Olga Kozlova, Elena Vladimirtseva, Svetlana Smirnova, Anna Lipina and Olga Odintsova for publication in the Coatings journal.

The manuscript provides valuable insights into the new type of materials composed of textiles coated with microcapsules containing bioactive agents and testing them as materials for medical applications.

I reviewed this article some time ago, and I suggested a few changes in the manuscript before its publication. Now I see that the Authors took into account my suggestions. Also, the comments other reviewers; and the revised version of the manuscript looks pretty well. The manuscript seems to be undoubtedly better.

Comments:

  1. Although the Authors agreed with me that the name of reducing agent oxalic acid dialdehyde is not correct, the manuscript begins from the incorrect name of this compound. (Abstract)
  2. The references are not entirely correctly cited; there are mistakes and lacks in descriptions. Authors should check all positions to cite the literature correctly in a form allowing us to find them easily.

Author Response

Dear Reviewer!

We are very grateful for your attention to our work. The error was found and corrected, line 8, references were reformed in accordance with the requirements of the journal.

Thank you for recommending the article for publication and for helping to improve its quality.

Reviewer 3 Report

Thank you for taking all suggestions into the account. 

Author Response

Dear Reviewer!

We are very grateful for your attention to our work. Thank you for recommending the article for publication and for helping to improve its quality.